# UniComposer: Band-Level Music Composition with Symbolic and Audio Unification

## Abstract

Multi-track deep music generation has largely focused on pre-specified structures and instruments. However, it remains a challenge to generate "band-level" full-length music that is capable of allocating instruments based on musical features, their expressive potential, and their performance characteristics differences. Moreover, the representations of symbolic and audio music have been treated as distinct sub-areas, without a unified architecture to join their own advantages. In this work, we introduce **UniComposer** [1], a novel music generation pipeline that composes at the band level, utilizing a hierarchical multi-track music representation complemented by four cascaded diffusion models which progressively generate rhythm features, and unified features extracted from both symbolic and audio music by autoencoders. Experiments and analysis demonstrate that UniComposer achieves a unified latent space for symbolic and audio music, and is capable of generating band-level compositions with well structured multi-track arrangements, surpassing previous methods in performances.

## 1 Introduction

The area of deep generative models for symbolic music (Wang et al. (2024), Hsiao et al. (2021), Huang & Yang (2020), among others) have witnessed advancements in a range of fronts. However, the approaches in the literature do not yet achieve band-level music generation. Band-level music has two critical features: 1) Instruments in a band collaborate to fulfill specific roles, such as harmony, rhythm, or accompaniment, which can differ from their roles in solo performances—a nuance overlooked in current multi-track generation. For instance, the piano often provides repetitive harmonies in a band, contrasting with its more varied melodies in solo settings. 2) Composers carefully select instruments to match the melody's characteristics, ensuring timbre, range, and expressiveness align with the intended emotion. They also arrange instruments dynamically, adjusting prominence based on sections or rhythmic changes. Current methods either replicate input instruments (Liu et al. (2022), Dong et al. (2018)) or rely on pre-defined user input (Dong et al. (2023), von Rütte et al. (2023), OpenAI (2024)).

For music data that take Audio formats such as MP3 (mpgedit (2003)) and WAV (bitsforbyte (2021)), there are billions of publicly accessible music tracks spanning a wide range of styles, instruments, and time periods. However, the situation is different for symbolic music. On one hand, symbolic-format music data are not as ample as the audio-format. On the other hand, methods for analysis and generation in symbolic music are often more structured and fine-grained, allowing explicit incorporation of music theory into the model architecture (Wang et al. (2024), von Rütte et al. (2023)). This enhances the model's ability to implicitly capture complex musical relationships. In contrast, methodologies for analyzing and generating audio music tend to rely solely on the model's inherent capacity to learn these relationships (Ji et al. (2023), Huang et al. (2023)), without explicitly incorporating music structure into the architecture design. However, no public work has yet developed approaches to organically combine the two forms of music, leaving this area largely unexplored.

Our work, UniComposer, simplifies band-level music modeling by introducing a hierarchical structure that groups instruments into monophonic, polyphonic, and percussion categories. Monophonic instruments provide chordal foundations, polyphonic instruments add melodic complexity, and percussion drives the rhythm. We use a two-step generation process: first, a basic rhythm is created

---

[1]Demo page is on: https://sites.google.com/view/unicomposer

for each type, then details and variations are added. This method is powered by four cascaded Transformer-based diffusion models, operating at the bar level unified feature space for symbolic music and corresponding time for single bar for audio music.

UniComposer combines the strengths of symbolic and audio music formats through two key methods. First, it converts audio into symbolic format, utilizing the structured nature of symbolic music for generation, grounded in music theory. Second, for symbolic music generation, it uses a bar decoder to extract notes from audio data, enhancing the model's performance through data augmentation. This is achieved via encoders respectively for symbolic and audio inputs, sharing a unified bar decoder. Both inputs are mapped to a common latent space, seamlessly integrating audio and symbolic music generation. Figure 1 illustrates the architecture.

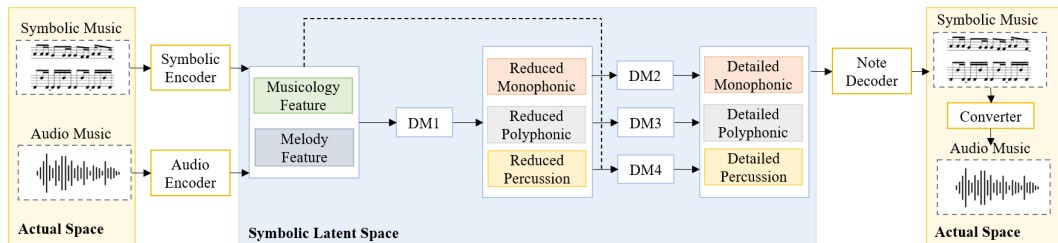

Figure 1: Overall Architecture of UniComposer.

In summary, the contribution of the paper is as follows:

1. We introduce UniComposer, the first band-level music generation system that involves collaborative roles of instruments, tailored to harmonize, provide rhythm, or accompaniment, with careful selection of instruments to match the melody's expressive qualities. Unicomposer uses a hierarchical representation across three instrument types.

2. We propose an approach that integrates the advantages of both symbolic and audio music, using separate encoders and a shared decoder to bridge both formats within a common feature space. This approach can leverage the rich audio music to facilitate data augmentation for symbolic music.

3. UniComposer is capable of mapping inputs into a unified feature space for generation through cascaded bar-based diffusion models. The composer can accept both symbolic and audio music as input, producing well-structured band-level music.

## 2 RELATED WORK

In this section, we first review multi-track music generation, followed by integrated processing for symbolic and audio music. Then, we review structured music modeling. Finally, we review the applications of autoencoders and diffusion models in music generation.

**Multi-Track Music Generation**. Addressing the complexity of generating music consisting of multiple interrelated tracks, multi-track music generation has become a significant area of research. Some works focus on developing multi-track representations of music which facilitate the generation process, while others aim to utilize multi-track approaches to improve music resolution and achieve fine-grained control. Guo (2024) integrates the AC algorithm to enhance diversity in multi-track music generation. MMBert (Zhu et al. (2024)) is a multi-track generation method focusing on the chord aspects of music. Jen-1 Composer (Yao et al. (2023)) generates music that contains four tracks from text prompts. Multitrack Music LDM (Karchkhadze et al. (2024)) leverages multi-track techniques to generate high-fidelity audio music. Cosenza et al. (2023) utilizes a graph-based generation approach to produce polyphonic music, with the instruments to be played specified by the user. MMMachine (Ens & Pasquier (2020)) mainly concentrates on multi-track inputs for music representation to achieve a better understanding of music, similar to SymphonyNet (Liu et al. (2022)) that copies input tracks for generation.

**Integrated Processing of Symbolic and Audio Music**. Research that combines audio and symbolic music is currently divided into two interconnected areas: audio rendering of symbolic music and automatic music transcription that translates audio into symbolic notes. For audio rendering of

symbolic music, traditional audio renderers use DSP-based or sample-based synthesizers to produce instrumental sounds. A parallel line of research has applied data-driven approaches to audio synthesis, often referred to as "neural audio synthesis." Notable examples include NSynth (Engel et al. (2017)), GANSynth (Engel et al. (2019)), DDSP (Engel et al. (2020)), and MIDI-DDSP (Wu et al. (2021)). These approaches share a similar concept of converting notes to audio waveforms using neural networks that perform an upsampling process.

For automatic music transcription, the outputs are typically frame-level multi-pitch estimation (MPE) or note-level estimation. These methods commonly first estimate a pitch posteriorgram, where each time-frequency bin is assigned an estimate of the likelihood of a fundamental frequency being active at a given time (Duan et al. (2010); Bittner et al. (2017)). Multiple methods have been proposed for estimating notes from pitch posteriorgrams, such as using median filtering (Klapuri & Davy (2007)), Hidden Markov Models (Benetos (2017)), or neural networks (Ewert & Sandler (2017); Nishikimi et al. (2021)). Gardner et al. (2021) is the first lighting work of leveraging a unified framework for transcription.

**Structured Music Modeling**. The local musical structure can be effectively modeled in two distinct ways: through approaches that either learn structure indirectly or through methods that directly define or extract musical elements, both yielding good results. Approaches like Music Transformer (Huang et al. (2018)), JukeBox (Dhariwal et al. (2020)), MuseFormer (Yu et al. (2022)), MERT (Li et al. (2023)), LLark (Gardner et al. (2023)) and Lemercier et al. (2024) represent the former, where models predict and generate musical events by capturing dependencies within the music. On the other hand, methods that rely on musical domain knowledge define specific features or extract interpretable musical representations, enabling the model to learn structures such as pitch contours at the measure level and accompaniment patterns. Some studies also aim to combine them, yielding high quality output (Mariani et al. (2023), Wang et al. (2024)).

**Autoencoders and Diffusion Models for Music Generation**. (Variational) Autoencoders are widely utilized to extract musical representations from pre-trained models, encapsulating both theoretical and perceptual aspects of music (Brunner et al. (2018); Jiang et al. (2020); Wu & Yang (2023)). These extracted features serve as essential tools for modeling cascading musical events due to their rich informational content. In the domain of music generation using diffusion models, two principal approaches have emerged. The first approach represents music in a piano-roll format, enabling diffusion models to directly generate note pitches and durations without the need for intermediate representations (Mittal et al. (2021), Atassi (2023), Wang et al. (2024)). The second approach employs diffusion models as feature learners; these models generate music by manipulating features encoded from preceding stages and executing reverse diffusion processes to synthesize the final output (Zhu et al. (2022), Zhang et al. (2023), Huang et al. (2024)).

## 3 METHODOLOGY

Given an input melody, our proposed UniComposer extracts features from each bar in symbolic music or the relative duration in audio. The composition process operates within the latent space of these features, categorized into three main functions: monophonic, polyphonic, and percussion. Feature extraction is managed by symbolic and audio encoders, while a shared decoder reconstructs bars from these extracted features. Using bars as the fundamental unit, the system progressively generates detailed features for each category through four cascaded diffusion models. A converter can subsequently translate symbolic output into audio. Section 3.1 outlines the hierarchical representation of band-level music, followed by the feature extraction together with the training strategy in Section 3.2, 3.3 and 3.4. Finally, the overall generation process is discussed in Section 3.5.

### 3.1 HIERARCHICAL REPRESENTATION OF BAND-LEVEL MUSIC

To streamline and simplify the modeling of band-level music, we adopt a functional separation approach to represent the three components of the musical ensemble, with a hierarchical reduction process transitioning from detailed to simplified rhythmic structures inspired by Prajwal et al. (2024) and Wang et al. (2024) to capture the overarching compositional framework. This approach enhances the efficiency of the learning process for UniComposer, and is illustrated in Figure 2.

**Functional Separation**. Monophonic instruments play one note at a time, while polyphonic instruments can produce multiple notes simultaneously, enabling harmonic richness. Percussion instruments create sound by actions like striking or shaking. Monophonic instruments focus on melody, contributing to musical expression by adding emotional depth through greater variation, while polyphonic ones often emphasize harmony and accompaniment with repeated or transposed notes, supporting the music's structure. Percussion maintains rhythm, articulating the beats and tension that drive the progression of a piece. Recognizing these functional distinctions, UniComposer separates these musical elements to improve data representation for band-level music. This division is facilitated by referencing the program IDs on the MIDI standard (MIDI-Association (2024)).

**Hierarchical Reduction**. On one hand, as is typical for composers, the initial step often involves drafting each section of the ensemble, defining key structural elements such as the primary modulation points and the bassline pitches. On the other hand, the process of function separation can result in superimposition of different instrumental parts, leading to redundant and overly intricate note sequences, which makes it challenging for UniComposer to learn. Inspired by Wang et al. (2024), we adopt a similar reduction approach based on the detailed rhythmic data for each instrumental category. The rhythmic structure of both monophonic and polyphonic sections is largely influenced by the harmonic chroma, while the rhythm of percussive instruments is primarily determined by the time signature, as it governs the temporal shifts within a composition.

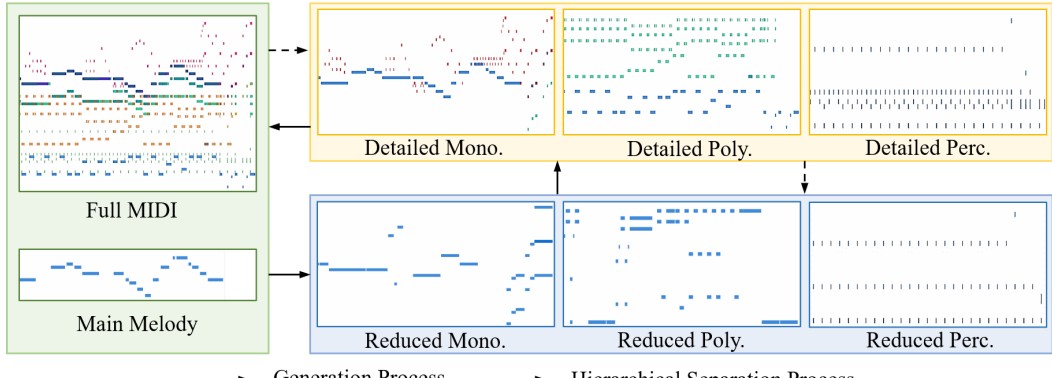

Figure 2: Band-Level Hierarchical Music Representation. Green box shows music with pre-labeled main melody. Mono., Poly. and Perc. represent monophonic, polyphonic and percussion, respectively. Hierarchical separation process breaks down band-level music into three parts, from detailed to reduced. Generation process starts with the main melody and adds reduced and detailed components based on instrument separation.

### 3.2 BAR-LEVEL FEATURE EXTRACTION FOR SYMBOLIC MUSIC

To analyze a sequence of notes within a bar, UniComposer extracts two key features: one representing the notes themselves (note feature) and the other capturing the overall musical characteristics of the bar (musical feature). This process is facilitated by the joint training of a symbolic encoder and a bar decoder, inspired by the approach outlined in Roberts et al. (2018), but with a more lightweight model that utilizes Transformer architecture (Vaswani (2017)) instead of RNNs.

**Data Representation**. UniComposer employs a modified version of the REMI representation (Huang & Yang (2020)), specifically adapted to handle bar-level features in symbolic music. Four key attributes are extracted for each bar: chord, time signature, dynamics, and tonality. Chords are represented by their root, chroma, and bass, while time signatures are restricted to three discrete values: 3/4, 6/8, and 4/4. Dynamics are categorized into six levels, ranging from pianissimo (pp) to fortissimo (ff). Tonality, indicating the organization of chords, can be either major or minor. Each note is characterized by five attributes: onset, duration, pitch, velocity, and instrument. To achieve a fine-grained temporal resolution, each bar is divided into 24 time bins. Velocity is simplified to four levels (von Rütte et al. (2023)), preserving essential nuances while reducing complexity. Pitch is encoded using 128 discrete values (MIDI-Association (2024)), and We group the 128 MIDI instruments, along with drums, into 13 categories, such as classifying all types of pianos (e.g., acoustic

piano, electric piano) under the piano category. UniComposer compiles all the notes present in its dataset and uses them (which is approximately $10^6$) as its vocabulary.

**Addressing Embedding Challenge**. Embedding layers trained alongside the model (e.g., Liu et al. (2022), von Rütte et al. (2023)) face instability due to the excessively large vocabulary size of Uni-Composer. Additionally, for this five-attribute note representation, the meaning of each note is inherently determined by its own attributes, contrasting with the word embedding approach (e.g., Mikolov (2013)), where meaning is derived from contextual co-occurrence. To tackle these challenges, UniComposer proposes an innovative solution: a simple multi-layer perceptron (E-MLP) is trained to encode each note by applying basic addition and concatenation operations to the traditional embedding vectors of the five attributes of a note. This process is followed by feed-forward layers, which produce an output embedding vector with a fixed dimensionality of 16.

**Symbolic Encoder**. The symbolic encoder employs the E-MLP mentioned above to generate a sequence of embedding vectors for the notes within a bar. Two special learnable vectors, representing the start $[BOS]$ and end $[EOS]$ of the sequence, are added to the embedding sequence. This sequence is then processed by a bi-directional model based on the classical multi-head transformer architecture proposed by Vaswani (2017), consisting of 4 layers and 4 attention heads is used to handle this embedding sequence. The transformer's final hidden state is passed through two separate linear projection layers, producing outputs for both note and musical feature representations.

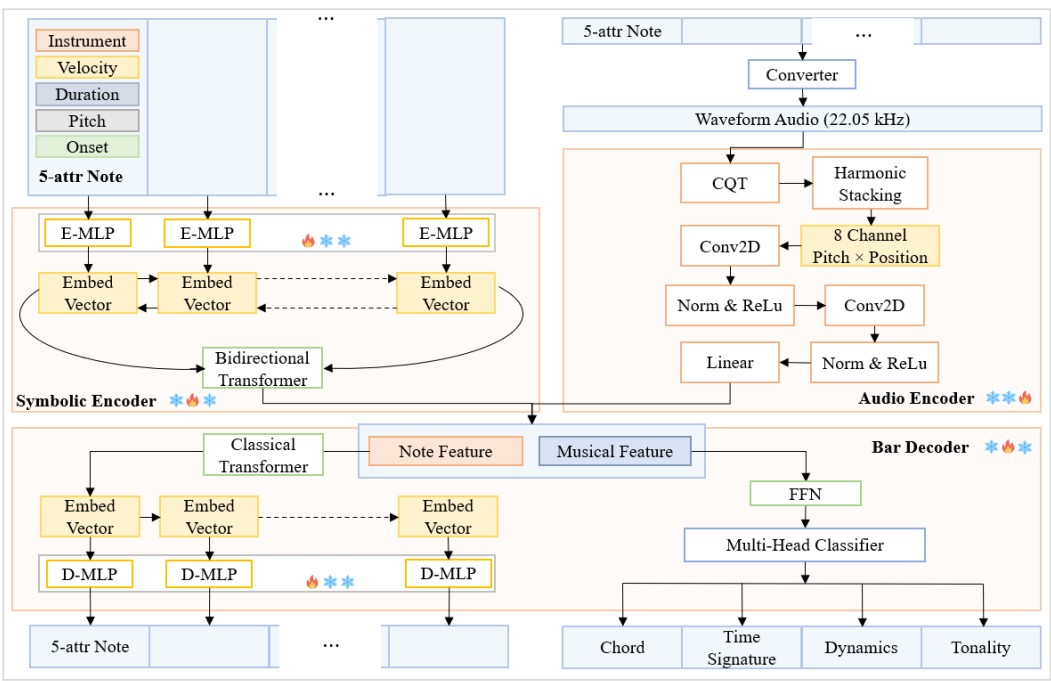

Figure 3: Overall Architecture for feature extraction by the encoders. The symbolic encoder and audio encoder process these two types of input, respectively, while a bar decoder is employed to reconstruct notes and predict musical features. The flame and snowflake icons in the image correspond to Table 1, indicating the three-step training schedule for each module.

### 3.3 UNIFICATION OF AUDIO INTO SYMBOLIC MUSIC

**Pre-processing**. Since general audio formats such as MP3 and WAV are dense time-sequence data, UniComposer applies the Harmonic Constant-Q Transform (H-CQT) (Bittner et al. (2017)) to process them. H-CQT takes an audio signal as input, and the output is a three-dimensional tensor representing the time-frequency representation at different harmonic frequencies. Its main function is to capture both the fundamental frequency and its harmonic components, providing a richer spectral representation. UniComposer computes H-CQT with 3 bins per semitone and a hop size of 11 ms, operating with a sample rate of 22.05 kHz and utilizes 8 harmonics. For an input of shape $(1, t \times 22,050)$, this H-CQT produces an output with the shape $(8, t \times 172, 128)$.

**Audio Encoder**. The audio encoder aims to map audio segmentation to the same note and musical feature space of the symbolic encoder. After pre-processing, a fully convolutional model followed by an output layer is applied to produce the note and musical feature. The architecture is similar to that proposed by Bittner et al. (2022), but with fewer layers, as the audio encoder processes shorter input segments. Since H-CQT captures all harmonic waves, the convolutional layers with deep kernels are able to detect notes with correct pitch and duration. Finally, using a series of linear and activation layers, the audio encoder generates the expected two features.

## 3.4 Refactoring and Training Process for Feature Extraction

**Bar Decoder**. The task of the bar decoder is to reconstruct all the notes from the input based on the note feature, and predict four musical attributes of the bar based on the musical feature. For note reconstruction, the bar decoder uses a classical transformer decoder with 4 layers and 4 attention heads. The note feature is replicated $n$ times to serve as the input hidden state of the decoder. The transformer generates an embedding vector for each position auto-regressively, with a dimension of 16. For musical feature prediction, a multi-layer percepton is applied to the musical feature, followed by four classifiers to predict the four musical attribute of a bar.

**Three-Step Training Process**. Firstly, the E-MLP is optimized to establish a stable embedding space for the notes. A separate MLP architecture (D-MLP), which includes five distinct classifiers, is employed to decode specific attributes from the E-MLP's output. Secondly, the symbolic encoder and bar decoder are trained jointly in a self-reconstruction manner for symbolic music. Specifically, the symbolic encoder extracts both the note and feature from a musical bar, while the bar decoder attempts to reconstruct the bar and predict its musical attributes. Thirdly, the bar decoder is frozen, and a similar self-reconstruction training approach is applied to the audio encoder, in conjunction with the bar decoder. The key difference here is that a converter, based on the open-source tool Fluidsynth (Tom Moebert (2024)), is used to convert symbolic music into audio waveforms, which serve as the input for the audio encoder. These three steps can be found in Table 1.

Table 1: Three step training process for the encoders

| Step | Trainable Part | Target |
|---|---|---|
| 1 | E-MLP, D-MLP | Stable Embedding Vector |
| 2 | Symbolic Encoder, Bar Decoder | Stable Symbolic Feature Space |
| 3 | Audio Encoder | Unified Audio and Symbolic Feature |

## 3.5 Hierarchical Generation in Feature Space

UniComposer operates following the process depicted by the solid lines in Figure 2, but within a unified feature space. The generation process begins with the main melody's features, progressing from reduced to detailed. Finally, the feature are decoded and merged together.

**Cascaded Diffusion Models**. UniComposer utilizes four Transformer-based diffusion models (denoted as DM1 to DM4), all of which share the same architecture. They work within the bar-level unified feature space, using background conditions to generate target features from Gaussian noise. Starting with the note and musical features extracted from a given melody, DM1 generates the reduced monophonic, polyphonic, and percussion features for each bar. Then, with the note and musical features, combined with one of the reduced features as part of the background condition, DM2–DM4 generate detailed features for each bar. Detailed configurations for every diffusion model can be found in Table 2.

Table 2: Configuration of the four diffusion models. 'R-' stands for 'Reduced-'. $n$ represents the number of bars, and each channel has the shape $(n, 256)$, as each type of feature is represented by 256 dimensions.

| | Background Cond. | | Output | |
|---|---|---|---|---|
| | Shape | Interpretation | Shape | Interpretation |
| DM1 | $(2, n, 256)$ | Note, Musical | $(3, n, 256)$ | R-Mono., R-Poly., R-Perc. |
| DM2 | $(3, n, 256)$ | Note, Musical, R-Mono. | $(1, n, 256)$ | Detailed Mono. |
| DM3 | $(3, n, 256)$ | Note, Musical, R-Poly. | $(1, n, 256)$ | Detailed Poly. |
| DM4 | $(3, n, 256)$ | Note, Musical, R-Perc. | $(1, n, 256)$ | Detailed Perc. |

**Attention Mechanism**. Inspired by Shabani et al. (2023), we design three types of attention mechanisms. 1) Self Attention, which restricts the attention to within a single bar to enhance local information; 2) Global Attention, a standard self-attention applied across all bars in a song; and 3) Local Attention, which operates at a 4-bar level to help the model capture short-term dependencies within the song. In each attention layer, we learn three separate sets of key, query and value matrices, using four multi-heads. The outputs of the three attention mechanisms are summed and then passed through a standard Add & Norm layer. The denoising process repeats this attention block six times. The masks used to implement these three types of attention mechanisms are shown in Figure 4.

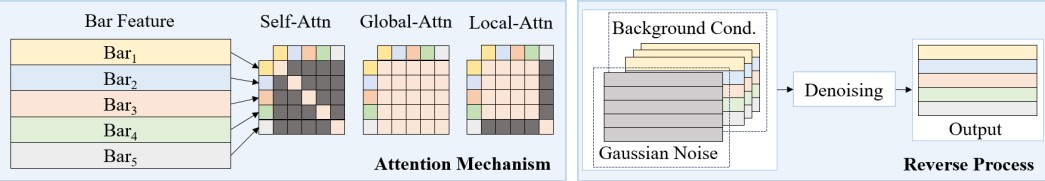

Figure 4: Attention mechanism and diffusion details. Three types of attention are applied using masks and separate query, key and value sets. The Transformer-based diffusion model learns to reconstruct the features of each bar from Gaussian noise, based on specific background conditions.

**Data Augmentation Derived from Audio**. Due to the richness of audio data, this can be achieved in two ways: 1) For audio tracks that have already been separated by instrument type, which is common in DJ music, both the input and output of DM1 can be derived by combining tracks based on their track names, utilizing features extracted from the audio encoder. 2) For general audio music, the combination of the audio encoder and bar decoder can be used as an automatic music transcription system, converting audio into symbolic music, thereby enhancing data diversity for training.

## 4 EXPERIMENTS

We implemented UniComposer using PyTorch, with the diffusion models built upon a public implementation of Guided-Diffusion (Dhariwal & Nichol (2021)). All experiments were conducted on a single NVIDIA RTX 4090 GPU. In the following sections, we first introduce the composition of the dataset used to train each part of UniComposer, and then evaluate its performance in terms of the unified feature and the generated quality. We also conduct ablation studies on the attention mechanism and the cascaded diffusion models, demonstrating their necessity.

### 4.1 DATASET

We use notes, bars, and songs from the Lakh MIDI Dataset (LMD, Raffel (2016)) to train UniComposer. The LMD dataset contains over 170,000 unique multi-track MIDI files, with each track labeled by instrument name. We reserve 1,000 songs as evaluation set.

**Dataset for encoders**. In Step 1, we collect unique notes from LMD to train the E-MLP and D-MLP. In Step 2, we use non-empty bars with varying note densities, instruments, and durations to train the symbolic encoder and bar decoder. In Step 3, we first synthesize entire songs from MIDI files using Fluidsynth (Tom Moebert (2024)), then extract the corresponding segments of bars filtered in Step 2 from the audio. This process results in MIDI-audio bar pairs, which are used to train the audio encoder while keeping the bar decoder frozen.

**Dataset for Cascaded Diffusion Models**. For training the four diffusion models, we filter out songs in LMD that are either too short or too long, selecting only those that contain a pre-labeled "melody" or "main melody" along with at least one monophonic instrument, one polyphonic instrument, and one percussion instrument. A detailed breakdown of the training dataset is provided in Table 3.

Table 3: Dataset composition for encoder, decoder and cascaded diffusion models of UniComposer

| Part | Encoder and Decoders | | | Cascaded Diffusion Models | | | |
|---|---|---|---|---|---|---|---|
| | Step1 | Step2 | Step3 | DM1 | DM2 | DM3 | DM4 |
| Unit | Note | Bar | MIDI-Audio Bar Pair | Song | Song | Song | Song |
| Count | 608,020 | 915,644 | 915,644 | 13,232 | 13,232 | 13,232 | 13,232 |

## 4.2 Evaluation on Unified Feature

**Method and Metrics**. To assess the performance of our encoders, we focused on the note estimation task by integrating the audio encoder and bar decoder (A&B). This system takes audio input and estimates notes, reflecting the core capabilities of the model. Performance is measured using the note-level F-measure ($F$), where a note is considered correct if its pitch, onset, and offset fall within a defined threshold compared to the ground truth. Additionally, we report the note-level F-measure without considering offsets ($F_{no}$), following the same criteria as the F-measure but ignoring offsets, as well as the bar-level note accuracy ($Acc$). These metrics are computed using the mir_eval library (Liang et al. (2015)).

**Baseline Settings**. As a baseline, we compare the performance of A&B on note onset and duration with other note estimation pipelines, including Basic-Pitch (Bittner et al. (2022)) and MI-AMT (Wu et al. (2020)). To assess the necessity of E-MLP and D-MLP, we also create a variation of Uni-Composer where these components are replaced by a vocabulary of 608,020 tokens, followed by a standard embedding layer. This model is denoted as Vocab-AE. The reserved evaluation set are separated into main melody, monophonic and polyphonic tracks, and we report the note reconstruction accuracy in Table 4.

**Analysis**. It is important to note that Basic-Pitch and MI-AMT only estimate pitch, onset, and duration, whereas A&B and Vocab-AE also predicts velocity and instrument. Vocab-AE performs worse, emphasizing the value of the stable note embeddings produced by E-MLP. A&B performs on par with Basic-Pitch in most cases. Given that A&B operates in a unified feature space while Basic-Pitch is specialized for note estimation, A&B demonstrates strong performance.

Table 4: Note estimation evaluation of Audio Encoder & Bar Decoder (A&B) with other baselines

|  | Melody | | | Mono. | | | Poly. | | |
|---|---|---|---|---|---|---|---|---|---|
|  | $Acc$ | $F_{no}$ | $F$ | $Acc$ | $F_{no}$ | $F$ | $Acc$ | $F_{no}$ | $F$ |
| **Vocab-AE** | 0.44 | 0.36 | 0.31 | 0.12 | 0.08 | 0.04 | 0.11 | 0.06 | 0.02 |
| **Basic-Pitch** | 0.91 | 0.79 | 0.71 | 0.76 | 0.69 | 0.61 | 0.70 | 0.65 | 0.63 |
| **MI-AMT** | 0.80 | 0.68 | 0.62 | 0.64 | 0.56 | 0.50 | 0.42 | 0.34 | 0.25 |
| **A&B** (Ours) | 0.94 | 0.82 | 0.74 | 0.72 | 0.63 | 0.57 | 0.77 | 0.62 | 0.59 |

## 4.3 Evaluation on Generation Quality

**Metrics**. A variety of metrics have been proposed to evaluate the harmony, quality and similarity of music generation. We borrowed the metrics from Ji et al. (2023) and Ren et al. (2020) for evaluation: chord accuracy (CA) that measures the harmony, and averaging overlapped area (OA) of distributions ($\mathcal{D}_\mathcal{A}$, $\mathcal{A}$ can be one of $P(itch)$, $V(elocity)$, $D(uration)$, and $OI(onsetInterval)$) to measure the difference between generated musical piece and ground-truth musical piece.

**Baseline Settings**. We introduce MuseGAN (M-G, Dong et al. (2018)), PopMAG (P-M, Ren et al. (2020)) and Figaro (F-G, von Rütte et al. (2023)). To make UniCompoer (U-C) comparable with the three model: 1) For MuseGAN, we take 4 bars as input to generate 4 tracks (guitar, drum, string and bass) conditioned on a main melody of piano track, and free of musical information since MuseGAN doesn't use chord and other features. 2) For PopMAG, we use the same task, while extend the generation length into 64 bars. 3) For Figaro, we use only features including chord, beat and target instrument as input, as Figaro just take them as input to generate new rhythm. We compare the ability of UniComposer with them separately on evaluation set. Results are in Table 5.

Table 5: Music Geneartion Quality Evaluation. The three groups carry out different tasks.

|  | $CA$ | $\mathcal{D}_\mathcal{P}$ | $\mathcal{D}_\mathcal{V}$ | $\mathcal{D}_\mathcal{D}$ | $\mathcal{D}_{\mathcal{OI}}$ |
|---|---|---|---|---|---|
| **M-G** | $0.334 \pm 0.011$ | $0.294 \pm 0.013$ | $0.342 \pm 0.008$ | $0.311 \pm 0.016$ | $0.336 \pm 0.010$ |
| **U-C** | $0.397 \pm 0.010$ | $0.358 \pm 0.012$ | $0.407 \pm 0.016$ | $0.301 \pm 0.009$ | $0.398 \pm 0.013$ |
| **P-M** | $0.589 \pm 0.011$ | $0.601 \pm 0.013$ | $0.492 \pm 0.009$ | $0.523 \pm 0.007$ | $0.655 \pm 0.007$ |
| **U-C** | $0.566 \pm 0.010$ | $0.625 \pm 0.018$ | $0.511 \pm 0.014$ | $0.507 \pm 0.005$ | $0.628 \pm 0.014$ |
| **F-G** | $0.541 \pm 0.011$ | $0.356 \pm 0.006$ | $0.642 \pm 0.008$ | $0.477 \pm 0.008$ | $0.433 \pm 0.004$ |
| **U-C** | $0.445 \pm 0.012$ | $0.451 \pm 0.016$ | $0.592 \pm 0.011$ | $0.504 \pm 0.011$ | $0.507 \pm 0.008$ |

**Instrument Assignment Capability**. Since UniComposer is the first system to automatically select instruments, metrics are limited. To assess its effectiveness, we analyzed the instrument distribution across 10 melodies in four emotional categories: happy, sad, soothing, and stirring. The occurrence of guitar, violin, trumpet, and flute in these melodies was recorded and summarized in Table 6.

Table 6: Instrument occurrence in different emotional melodies, each category containing 10 pieces.

|  | Guitar | Violin | Trumpet | Flute |  | Guitar | Violin | Trumpet | Flute |
|---|---|---|---|---|---|---|---|---|---|
| **Happy** | 8 | 5 | 3 | 0 | **Soothing** | 7 | 7 | 2 | 1 |
| **Sad** | 8 | 7 | 0 | 0 | **Stirring** | 8 | 6 | 3 | 0 |

## 4.4 Ablation Study

**Attention Mechanisms**. We begin by experimenting with the global attention mask (U-GA), a common approach in natural language processing. Next, we test the addition of either the self-attention mask (U-GSA) or the local attention mask (U-GLA) to global attention individually. Finally, we compare these three attention mechanisms with UniComposer (UniComp.). As shown in Table 7, global attention establishes a foundation by providing cross-bar focus within a music piece. Self-attention refines this by encouraging the model to emphasize pitch and velocity, yielding slight improvements. Local attention, which mimics a composer's focus on musical sub-structures, further enhances overall quality.

**Cascaded Diffusion Models**. To evaluate the hierarchical generation process, we first removed diffusion models DM2 through DM4, leaving only DM1 (U-DMa). In this configuration, DM1 is trained to generate the entire MIDI based solely on the melody and musical features. We also implemented a second structure in which DM2 through DM4 were compressed into a single diffusion model (U-DMb), while DM1 remained unchanged. In this variant, the monophonic, polyphonic, and percussion features were combined, and a single diffusion model was trained to generate the complete raw MIDI. As shown in Table 7, the results demonstrate that it is challenging for a single diffusion model to capture the full data distribution, from the main melody to the detailed instrumentation. Due to the complexity of each individual instrument, integrating them into a single diffusion model proves difficult.

Table 7: Ablation study on attention mechanisms and cascaded diffusion models.

|  | $CA$ | $\mathcal{D}_{\mathcal{P}}$ | $\mathcal{D}_{\mathcal{V}}$ | $\mathcal{D}_{\mathcal{D}}$ | $\mathcal{D}_{\mathcal{OI}}$ |
|---|---|---|---|---|---|
| **U-GA** | $0.549 \pm 0.008$ | $0.501 \pm 0.006$ | $0.405 \pm 0.004$ | $0.492 \pm 0.007$ | $0.540 \pm 0.006$ |
| **U-GSA** | $0.532 \pm 0.009$ | $0.530 \pm 0.006$ | $0.421 \pm 0.010$ | $0.493 \pm 0.006$ | $0.559 \pm 0.010$ |
| **U-GLA** | $0.601 \pm 0.008$ | $0.627 \pm 0.011$ | $0.537 \pm 0.012$ | $0.507 \pm 0.006$ | $0.629 \pm 0.015$ |
| **U-DMa** | $0.187 \pm 0.006$ | $0.156 \pm 0.002$ | $0.174 \pm 0.004$ | $0.102 \pm 0.001$ | $0.168 \pm 0.004$ |
| **U-DMb** | $0.364 \pm 0.008$ | $0.420 \pm 0.007$ | $0.386 \pm 0.009$ | $0.366 \pm 0.005$ | $0.309 \pm 0.004$ |
| **UniComp.** | $\mathbf{0.607 \pm 0.008}$ | $\mathbf{0.644 \pm 0.013}$ | $\mathbf{0.549 \pm 0.011}$ | $\mathbf{0.511 \pm 0.009}$ | $\mathbf{0.643 \pm 0.010}$ |

## 5 Conclusion

In this work, we propose UniComposer, a band-level music generation pipeline that involves collaborative roles of instruments and bridges the gap between symbolic and audio music. UniComposer first extracts features from both symbolic bars and audio segments, and then applies four cascaded diffusion models to generate the corresponding features for each bar. Experiments on the feature space demonstrate the effectiveness of the unified latent space, while comparisons with other music generation pipelines highlight UniComposer's ability to generate band-level music. Additionally, the ablation studies confirm the effectiveness of the various components within UniComposer.

**Discussion and Future work**. There are potential discrepancies in data distribution between synthesized audio generated from MIDI, background noise, and real-world recordings. However, a limitation is the lack of extensive datasets containing paired raw audio and corresponding MIDI for comprehensive training. Meanshile, UniComposer's capacity to process human vocals remains limited due to insufficient data. Another constraint is the requirement for prior BPM (Beats Per Minute) information, which can restrict the data. Addressing these issues may be a focus of future research.

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
