# OpenReview forum: "UniComposer: Band-Level Music Composition with Symbolic and Audio Unification"
_ICLR.cc/2025/Conference — Submitted to ICLR 2025_

### Official Review · Reviewer_BEEG · 2024-10-31

**Soundness:** 3
**Presentation:** 1
**Contribution:** 2
**Rating:** 3
**Confidence:** 4

**Summary:**

This paper introduces UniComposer, a diffusion-based music generation model targeting band-level, full-length music. It first introduces a hierarchical music representation from note-level attributes to bar-level encodings. On top of this, a series of diffusion models learn to generate bar encodings on the full-length level. To enhance musical structure comprehension, instruments are classified into three functional categories—monophonic, polyphonic, and percussive—each modeled by separate diffusion modules conditioned on shared melody and musical features. UniComposer also supports audio input for bar encodings, using a unified symbolic/audio latent space. Experimental results demonstrate generation quality based on objective metrics compared to baselines, with ablation studies validating the unique attention mechanism and category-specific diffusion modules.

**Strengths:**

* This paper presents a working scheme for hierarchical music modelling: from note-level attribute learning, via bar-level encoding, to song-level bar feature prediction. It further illustrates a scheduled training strategy, which realizes each of the three hierarchies effectively.

* By learning a unified audio/symbolic encoding space, the proposed model can also work as a transcriptor for AMT tasks, while instrument and note velocity are supported along with other note attributes.

**Weaknesses:**

* Weakness in experiment:

   1. Lack of ablation study on the added audio branch: While this paper aims to "join the advantage" of both sym/audio music, one may expect the added audio branch to actually benefit symbolic music generation (otherwise, the motivation for sym/audio unification seems less sufficient). Therefore, it could be necessary to have an ablation study on the model trained with and without the audio branch.
   2. Baseline models: A few alternative baseline models that may worth refering [1, 2]
   3. Evaluation of music quality: Evaluate music generation using statistical metrics is arguably sufficient. Human study might be necessary to test the naturalness, musicality, and creativity of the generated results.

* Weakness in writing:
   1. Clarity in the introduction part could be further improved. The current version states "music generation" broadly, while the specific task seems to be accompaniment generation based on the later part of the paper. In other words, let the readers know the input and output of the task in the first place. This can offer an expectation to help comprehend the whole passage. Another problem lies in Line 058 which states "converts audio into symbolic format." while the model training primarily synthesizes symbolic into audio.

   2. The division between Mono. and Poly. (Line 164)  might be a bit confusing because Bass, which is mostly monophonic, is apparently grouped into Poly. Maybe here Melodic and Harmonic are better wording to name the two categories.

   3. Line 308 introduces the cascaded diffusion models as "Transformer-based." If the reviewer understands correctly, it should actually be U-Net (convolutional) based with added self-attention modules. Note that this architecture has significant distinction from Transformers.

   4. In the experiment part ( Section 4.4), there is no interpretation on the evaluation results

* Demo page is not working


[1] H.-W. Dong et al. Multitrack Music Transformer, in ICASSP 2023.

[2] J Thickstun. Anticipatory Music Transformer. TMLR, 2024.

**Questions:**

* Does the UniComposer model support any user control over the generation process?

* How is the chord of a piece extracted (Line 208)? What is the chord resolution (i.e., #chord per bar)?

* What is the loss for the "Transformer-based" diffusion models (Line 308)? Is it MSE loss over the bar encodings?

* Could A&B (Section 4.2) compare to MT3 [1] on the multi-instrument music transcription task?

* Based on the reviewer's understanding, the Figaro model is a style transfer model, where symbolic features from piece A and audio features from piece B are required (as input) to generate a new piece. How is the Figaro model applied in this work as a baseline?

[1] J. Gardner, et al. MT3: Multi-task multitrack music transcription, in ICLR 2022.

---

> ### Author Response · Authors · 2024-11-20
>
> Thank you very much for your time and comments evaluating our work.
>
> ## Weakness
> **In Experiment**:
> 1. Our initial motivation for integrating audio into this framework was to create a more cohesive generation process, as audio and symbolic music inherently represent the same musical content. We have considered leveraging transcribed symbolic music from rich audio datasets to expand the training data. However, the current model has limited transcription capabilities (as demonstrated in the table provided later). As a result, in the current version of UniComposer, the exclusion of the audio component impacts functionality more significantly than performance.
>
> 2. Regarding the referred baseline models, we have conducted a comparison on the transcription task with [1] and plan to further include a reference to [2].
>
> 3. We acknowledge that the lack of subjective evaluation or a human study is a limitation of our current work due to constraints in time and resources, and we plan to address this in the future. Thank you for highlighting this important point.
>
> **In Writing**:
> 1. Thank you for pointing out the weakness in our writing. UniComposer is indeed a model designed to generate a piece of accompaniment given a piece of input. The phrasing in Line 058 was a mistake in writing. What we intended to convey was: "first mapping audio into the unified latent space with symbolic music.".
>
> 2. Regarding the division of instruments, we appreciate for highlighting the inaccuracy in wording. We appreciate your suggestion of using "melodic" and "harmonic" as descriptors, in fact we had previously considered them. Our intention was to emphasize that, whether multiple notes can be played simultaneously. Instruments like the piano can serve as both melodic and harmonic, depending on the context. We sincerely thank you for bringing this issue to our attention.
>
> 3. About the using of term "Transformer-based", we initially intended to express was "U-Net-based with added attention modules.". This was a mistake in writing, and thank you for figuring out this.
>
> 4. We are sorry for the oversight about the demo page, which was not successfully linked to the provided anonymous website. This annoymous link has been fixed [website](https://sites.google.com/view/unicomposer), which is now linked to the correct demo page. We much appreciate the reviewer's pointing out this issue to us.
>
> ## Questions
> **Q1**:  As shown in Fig. 3, the musical feature encapsulates information ranging from chords to tonality. While the note feature can be extracted directly from the input, the musical feature could instead be derived from a specific music piece with distinct styles and characteristics. This specific music piece could be treated as a form of control.
>
> **Q2**: To determine the chords, we first identify and register the chord, its chroma, and its bass for each chord (e.g., major C). For each bar, we calculate the cosine similarity between all pitches and all possible chord chromas. This approach allows us to assign a chord to every bar. The chord resolution in our method is at the bar level.
>
> **Q3**: The loss function used is the MSE loss, computed between the predicted feature and the ground truth feature for every bar encoding.
>
> **Q4**: Using the same evaluation metrics and the reserved evaluation set as described in Section 4.2, we present a comparison between A\&B and MT3 (mixture):
> | | $Acc$ | $F_{no}$ | $F$ | $Acc$ | $F_{no}$ | $F$ | $Acc$ | $F_{no}$ | $F$ |
> | -------- | -------- | -------- | -------- | -------- | -------- | -------- | -------- | -------- | -------- |
> | **MT3** | 0.90 | 0.86 | 0.80 | 0.85 | 0.79 | 0.71 | 0.82 | 0.74 | 0.68 |
> | **A\&B** | 0.94 | 0.82 | 0.74 | 0.72 | 0.63 | 0.57 | 0.77 | 0.62 | 0.59 |
>
> We acknowledge that MT3 outperforms A\&B in the multi-instrument transcription task. We believe this limitation is acceptable, as A\&B also maps symbolic and audio music into the same latent space, providing a unified representation that offers advantages in accompaniment generation.
>
> **Q5**: Figaro introduces the concept of using cascaded diffusion models to progressively add musical details, effectively managing the complexity of music generation. UniComposer adopts this idea but operates in a latent space, which distinguishes it from Figaro. Additionally, we would like to clarify that UniComposer generates features separately for symbolic and audio inputs. Rregardless of whether the input is symbolic or audio, the same features are extracted and processed using the same workflow.

---

### Official Review · Reviewer_K5Mz · 2024-11-01

**Soundness:** 2
**Presentation:** 1
**Contribution:** 1
**Rating:** 3
**Confidence:** 3

**Summary:**

This paper introduces a multi-track music composition algorithm containing three levels. The top level is a latent feature jointly learned from symbolic and audio sources; the middle level is music reductions of monophonic, polyphonic, and percussion tracks, and the bottom level is the actual generation. The model contains an autoencoder to learn the top-level features and cascaded diffusion models to generate the rest of the levels. Experiments show better objective metrics compared to the selected baseline.

**Strengths:**

Some of the design of this paper reflects the hierarchical planning involved in music generation. These are: 1) music is generated from "latent" ideas; 2) multi-track music can be categorized into different instrument types; and 3) generation is carried out from coarse-grained to fine-grained; for example, the authors leverage music reduction as an intermediate representation.

**Weaknesses:**

1. The motivation behind the proposed method could be clarified further. It would be helpful to understand the specific goals of the model---for example, whether it aims to enhance generation quality or address longer-term generation.
2. The demo page appears to be incomplete, which makes it challenging to assess the model’s capabilities and output quality fully. Additionally, there is no subjective evaluation. Objective metrics alone cannot provide a comprehensive understanding of the model's effectiveness.
3. The methodology is difficult to follow. The connections between graphs are unclear, as nodes are not consistently used across them, making it hard to trace the relationships. E.g., Where will the "Note feature" in Figure 3 be in other figures? Additionally, the section introduces many terms without effectively linking them, which obscures the overall approach.

**Questions:**

1. Could a completed demo page be provided to better assess the model’s capabilities?
2. It would be helpful to provide an explanation and add experiments on how features from symbolic music, audio, and the hierarchical design each contribute to the generation process.

---

> ### Author Response · Authors · 2024-11-20
>
> Thank you very much for your time and comments evaluating our work.
>
> ## Weaknesses
> **W1**. Our initial motivation was twofold:
> 1. To enable the **dynamic assignment of instruments**, moving beyond simply replicating input tracks or relying on pre-defined specifications.
> 2. To treat audio and symbolic music within a **unified framework**. These two points represent the marginal contributions of our work compared to the literature; this may haven't been sufficiently clarified in Section 2.
>
> **W2**. We are also sorry for the oversight that resulted in the incomplete demo page. The demo page was not successfully linked to the provided anonymous website. The current website link has been updated to be functional. Regarding subjective evaluation, we acknowledge this as a limitation of our current work due to constraints in time and resources, and we plan to address this in the future.
>
> **W3**. We regret any confusion caused by the wording and appreciate your attention and thoughtful comments. To clarify, the "Note Feature" and "Musical Feature" in Figure 3 correspond to the "Melody Feature" and "Musicology Feature" in Figure 1, respectively. Additionally, the "Bar Feature" in Figure 5 refers to all features in a single bar as described in Table 2.
>
> ## Questions
> **Q1**:  This anonymous link has been fixed [website](https://sites.google.com/view/unicomposer), which is now linked to the correct demo page. We much appreciate the reviewer's pointing out this issue to us.
>
> **Q2**: Allow me to provide a more detailed explanation:
>
> For both audio and symbolic inputs, the autoencoder extracts note features and musical features on a bar-by-bar basis. These extracted features are then processed by the cascaded diffusion models.
> To clarify, we considered the following aspects in our ablation studies:
> 1. **Note Feature**: This includes the five attributes of notes shown in the top-left corner of Figure 3. These attributes contribute significantly to the decoded notes.
> 2. **Musical Feature**: This encompasses high-level information illustrated in the bottom-right corner of Figure 3, serving as an auxiliary component for coherence.
> 3. **Cascaded Diffusion Models**: These models streamline the instrument-assignment task and reduce the reliance on an overly powerful single diffusion model.
>
> For (3), we conducted a brief ablation study as described in Section 4.4, with the results shown in **Table 7** (**U-DMa** and **U-DMb**). Regarding points (1) and (2), we have since added ablation studies to better illustrate their impact. We denote the UniComposer without the note feature as **w/o-N** and without the musical feature as **w/o-M**. Using the same evaluation metrics as described in Section 4.3, we obtained the following results:
>
>
> | | $CA$ | $D_P$ | $D_V$ | $D_D$ | $D_{OI}$ |
> | -------- | -------- | -------- | -------- | -------- | -------- |
> |**w/o-N**   | $0.544 \pm 0.011$ | $0.201 \pm 0.006$ | $0.342 \pm 0.008$ | $0.311 \pm 0.016$ | $0.374 \pm 0.007$ |
> |**w/o-M**  | $0.497 \pm 0.007 $ | $0.572 \pm 0.009$ | $0.507 \pm 0.016$ | $0.541 \pm 0.009 $ | $0.516 \pm 0.010 $ |
> |**U-C**  | $0.590 \pm 0.010 $ | $0.650 \pm 0.012$ | $0.588 \pm 0.006$ | $0.600 \pm 0.005 $ | $0.608 \pm 0.014$ |

---

> > ### Comment · Reviewer_K5Mz · 2024-11-27
> >
> > Thank you for the explanation and for finishing the demo page. These help readers better understand the idea.
> >
> > I still believe the paper lacks clarity. For example, I am still confused about why the "Reduced Poly" track on the demo page is monophonic and almost blank.
> >
> > I have listened to the generation results. My feeling is that the result is different from "full MIDIi" (a training example, I suppose), and the quality is lower than most existing models. Please correct me if I misunderstand.
> >
> > So, I will keep my score **unchanged**.

---

### Official Review · Reviewer_TpSi · 2024-11-03

**Soundness:** 2
**Presentation:** 2
**Contribution:** 1
**Rating:** 3
**Confidence:** 4

**Summary:**

The authors propose a complex system termed UniComposer to generate music in the MIDI multitrack format.
 This whole system is based on a bespoke representation of music and of the compositional process.
A MIDI multitrack is decomposed into Monophony/Polyphony/Percussion, each modality having a detailed and reduced version.
Training the whole system is done in 3 steps and involves training, among others, 4 diffusion models.
The main innovation seems to include an audio encoding encoder, concatenated to a (more classical) symbolic encoder.

**Strengths:**

A complex system to generate MIDI multitrack.

**Weaknesses:**

- With such a complex system, it is hard to understand the contribution. Differences with existing works are not clearly highlighted in the text.

- The most original contribution, which seems to consist in the addition of the audio is questionable:
The audio used as input of the audio encoder is obtained from a rendering of the MIDI file using FluidSynth (MIDI synthesizer). Such audio features are then concatenated to "symbolic features" obtained from the exact same part of the MIDI file. At the end, the model only predicts MIDI data (which are then rendered into audio).
In other words, there seems to be no additional information obtainable by considering these audio features.

- The accompanying website (last checked on 11/3/24) only features placeholder content
like
"Where we are today
What has your team accomplished? What are you most proud of? Tell site viewers some of your project's latest accomplishments.
Caption for a recent accomplishment
Caption for a recent accomplishment"

Overall, it seems like an overengineered system with no real novel insights, where the quality of the generations is impossible to evaluate.

**Questions:**

-

---

> ### Author Response · Authors · 2024-11-20
>
> Thank you very much for your time and comments evaluating our work.
>
> **Demo page**: We are sorry for the oversight about the demo page, which was not successfully connected to the provided anonymous website link. This anonymous link has been fixed [website](https://sites.google.com/view/unicomposer), which is now linked to the correct demo page. We much appreciate the reviewer's pointing out this issue to us.
>
> **Difference with existing works**: we hope the following points help clarify this issue:
> 1. Instrument Assignment: UniComposer introduces the capability to assign instruments dynamically, whereas existing works typically either replicate the input track or rely on pre-defined specifications for instrument names.
> 2. Unified Latent Space: By utilizing autoencoders, UniComposer maps both audio and symbolic music into a unified latent space, so that these two modes of data can jointly be used. This approach appears to be novel in the literature and may bring together the benefits of both audio (more data available) and symbolic music (fewer data, but more interpretable and controllable).
>
> **Use of FluidSynth-synthesized audio**: Our primary purpose was to address the issue of insufficient audio-MIDI pairs aligned at the bar level, which can be critical for training the autoencoder. We are not intended to extract some extra features from the audio. UniComposer is primarily trained on open-access MIDI data, along with audio data synthesized from these MIDI files. With its ability to map audio into MIDI, UniComposer can process real-world audio as input without requiring the training of a new model, offering greater flexibility and applicability.
>
> **Over engineered**: We appreciate the reviewer's comment about the over engineered system, which indeed makes the UniComposer system complex. We would like to provide some descriptions about the underlying thoughts. Our motivation was to design a system that leverages the advantages of both audio and symbolic music within a unified architecture. The autoencoder part needs to both facilitate the mapping of audio into MIDI and also to integrate discrete symbolic music into a single vector representation. This partly results in the complexity of the system. The cascaded hierarchical design of the four diffusion models, which decompose the complex task of music modeling, combined with autoencoders that transform the original sparse music representation into a dense format, makes UniComposer sophisticated yet hopefully more powerful and meanwhile, maintaining computational efficiency for class-aware, band-level music generation.

---

> > ### Comment · Reviewer_TpSi · 2024-11-25
> > **Thanks for the update**
> >
> > Thanks a lot for taking the time to answer + updating the website, which makes easier to grasp the results.
> > I raised my score for this, but still believes that there are more suitable conferences to showcase this work.

---

### Meta-Review · Area_Chair_fqLb · 2024-12-19

**Metareview:**

The paper presents a system for music generation using audio and symbolic data suggesting a unified representation, with further functional separation and hierarchical representation. The generation is done through four cascaded diffusion models which progressively
generate different musical features.

**Additional Comments On Reviewer Discussion:**

The reviewers noticed weaknesses in presentation and experimental results that do not merit publication at this time. Compared to baseline (Figaro) the proposed system (UniComposer) operates in a latent space. In addition, UniComposer offers a unified symbolic/audio latent space to allow for Audio input. During the rebuttal, the authors provided an ablation study and transcription performance comparison. Despite these correction, the reviewers noted that the paper is of insufficient quality both in terms of the experimental study, the demonstration examples and overall writing quality. Reviewer TpSi believes that there are more suitable conferences to showcase this work.  Reviewer K5Mz still believes that the paper lacks clarity, also noticing problems on the demo page.
The reviewer observes that the quality in the demo seems to be lower than most existing models, which seems to suggest a potential problem with the metric used in evaluation. The authors did not reply to these quality concerns after the first round of discussions.
Since the reviewers did not find the answers of the authors during the rebuttal round satisfactory, and unanimously did not recommend the paper for publication, I concur with this recommendation.

---

### Decision · Program_Chairs · 2025-01-22

Reject